# Experiencing Pregnancy during the COVID-19 Lockdown in Poland: A Cross-Sectional Study of the Mediating Effect of Resiliency on Prenatal Depression Symptoms

**DOI:** 10.3390/bs12100371

**Published:** 2022-09-29

**Authors:** Anna Studniczek, Karolina Kossakowska

**Affiliations:** 1Expert’s Antenatal School, St. Family’s Maternity Hospital in Warsaw, 02-544 Warsaw, Poland; 2Department of Clinical Psychology and Psychopathology, Institute of Psychology, Faculty of Educational Sciences, University of Lodz, Rodziny Scheiblerów Avenue 2, 90-128 Lodz, Poland

**Keywords:** prenatal depression, resilience, traumatic childbirth perception, pandemic-related pregnancy stress, COVID-19

## Abstract

The COVID-19 pandemic in Poland brought uncertainty, not only to the general population but also to women preparing for childbirth, which increased the risk of mental health illnesses during this special period of life. Resilience, which refers to positive adaptation or the ability to maintain good mental health, can be a protective factor against the development of psychiatric problems such as depressive symptoms. This study aimed to assess the protective role of resilience in the relationship of such risk factors as traumatic childbirth perception and pandemic-related pregnancy stress with prenatal depressive symptoms. The study was performed at the end of the first wave of the COVID-19 pandemic. A total of 80 pregnant women took part. A mediation analysis, an independent *t*-test, and a Pearson correlation analysis were conducted. The lower resilience group declared the inclusion of slightly more participants (*n* = 41; 51.2%); 39 women (48.8%) demonstrated a higher risk of prenatal depression. The analysis revealed a significant direct effect between pandemic-related stress and prenatal depression (βc = 0.285, SE = 0.05, t = 2.63, *p* < 0.05) as well as between pandemic-related stress and resilience (βa = −0.283, SE = 0.07, t = −2.61, *p* < 0.05) and between resilience and prenatal depression (βb = −0.585, SE = 0.07, t = −6.34, *p* < 0.001). After the introduction of resilience as a mediator, the strength of the relationship not only decreased, but also ceased to be statistically significant (βc′ = 0.120, SE = 0.04, t = 1.29, *p* = 0.19), which indicates that it was in a full mediation state (R^2^ = 0.39, F = 25.31, *p* < 0.001; Z = 2.43, *p* < 0.05). The results indicate that in pregnant women, a high level of resilience protects them from the effects of pandemic-related stress on perinatal depression symptoms.

## 1. Introduction

Pregnancy is a period of transformation for a woman, one that is characterized by a change of attitudes that requires her to take the role of a mother, especially when she is waiting for the birth of her first child. It is an extremely important stage in life for a woman and it is a period of anxiety [1] that is associated with an increased risk of depression [2]. Due to their situation, pregnant women constitute a special risk group that are burdened by greater concerns about their health and the health of the expected child. Previous studies have shown that some women fear childbirth to such an extent that they perceive it as a traumatic event; a strong fear of childbirth makes childbirth more difficult and prolongs the labor. In addition, contractions are perceived as stronger, and complications occur more frequently [3]. These types of difficulties can increase the risk of depression during pregnancy, and also, after childbirth [4].

The risk is greater when the external circumstances are particularly stressful, as was the case at the beginning of the COVID-19 pandemic. When the world media reported the growing number of new cases and deaths from COVID-19, it also had a significant impact on morbidity; a representative study of the Polish population showed that the severity of depressive symptoms during the first wave of the COVID-19 pandemic increased from 16.2% to 36.6% in the group of people that were aged 18 to 34 [5]. In rural areas, i.e., in those that are in Africa, the situation was even more difficult because of the lack of easy access to healthcare and the health worker shortage [6]. The disease spread out easily, also, because of a lack of knowledge among these rural communities. [6,7]. Older people that were living in rural communities in Japan were especially suffering from restrictions in their daily social lives [8,9]. The problems in these rural areas are complex, but the pandemic situation escalated them. In India, for example, people in isolation were left with no food and other basic amenities that are necessary for life [10].

Women who were pregnant during the first wave of the COVID-19 pandemic experienced the uncertainty of the future, the possibility of infecting themselves and their child with the coronavirus, and the inability of the child’s father to participate in the delivery [11,12,13]. Although they may have reacted adaptively to the stressors that were experienced during the COVID-19 pandemic, previous studies have shown that a large proportion of them react to stress with increased levels of negative emotions [14,15,16].

However, whether pregnant women react for various pregnancy-related situations in an adaptive or maladaptive way may result from their personal level of resilience. Resilience is defined as a universal ability that enables a person, group, or community to prevent, minimize, or overcome the harmful effects of an experienced misfortune [17], and as it is a multifactorial process of coping with unfavorable conditions that may develop, it therefore leads to their positive adaptation to them, during which, the individual, family, or extra-family protection factors reduce or compensate for the negative impact of the risk factors [18]. Resilience can, therefore, be understood as the ability to return to good functioning after a period of difficulties, losses, and stressors, and the ability to effectively cope with challenges or difficult situations. Indeed, coping with the expectations that are related to the knowledge about emerging risk factors, and coping well despite going through difficult experiences or recovering from trauma have been found to be related to mental resilience [19].

The role of resilience in women that are experiencing the perinatal period is the subject of a growing body of research. Some studies suggest that resilience can also play both a mediating [20,21] and a moderating [22] role between stress and anxiety symptoms and depression symptoms [22]. However, no studies have, so far, demonstrated that resilience has a protective role in the relationship between labor anxiety (and the fear of labor or a traumatic birth perception) and prenatal depression; although, admittedly, some studies have examined both of these variables, and the authors of these have analyze them in the opposite direction, i.e., their findings indicate that prenatal depression has a negative impact on the intensification of the symptoms of the fear of childbirth [23].

Since the outbreak of the coronavirus pandemic, we have seen the rise of factors that may have a negative impact on mental health in specific periods, such as the time that is spent waiting for a child to be born. While they are expecting a baby, women are particularly vulnerable to stressors, such as the fear of childbirth or pandemic-specific stressors. As such, it seems reasonable to ask whether, in such changed circumstances, resilience still has such a protective role. Thus, our study was designed to answer the following research questions:Is the level of prenatal depressive symptoms, pandemic-related pregnancy stress, and traumatic childbirth perception different among women with a low resilience when they are compared to those with high resilience?Were traumatic childbirth perceptions and pandemic-related pregnancy stresses related to depression symptoms among pregnant women during the COVID-19 lockdown?Does resilience act as a mediator in the relationship between a traumatic childbirth perception and prenatal depressive symptoms, and between pandemic-related pregnancy stress and prenatal depressive symptoms?

Based on the research questions that are mentioned above, we formulated the following hypotheses: (1) the levels of prenatal depression symptoms, pandemic-related pregnancy stress, and a traumatic childbirth perception are different among women with a low resilience when they are compared to those with high resilience; (2) a traumatic childbirth perception and pandemic-related pregnancy stress are independently associated with depression symptoms among pregnant women during the COVID-19 lockdown; (3) the associations between pandemic-related stress and prenatal depression, as well as a traumatic childbirth perception, are mediated through resiliency.

## 2. Materials and Methods

### 2.1. Study Design

This cross-sectional study was a part of a larger investigation to determine the relationship between resilience and selected psychological factors that are associated with the mental health issues that are experienced by pregnant women during the COVID-19 lockdown. The population was composed of Polish pregnant women. The primary outcome was the presence of depressive symptoms. The independent variables were a traumatic childbirth perception and pandemic-related pregnancy stress. Resilience was a mediator variable.

### 2.2. Ethical Consideration

The research procedure was performed following the Helsinki Declaration of Human Rights [24]. The study protocol was approved by the Committee for Bioethics of Scientific Research the Cardinal Stefan Wyszynski University in Warsaw. Participants were informed of the purpose, risks, and benefits of the survey and were told they could withdraw from the survey at any time, for any reason; all provided electronically informed consent to participate. The electronic informed consent form was prepared following the Ethics Guidelines for Internet-mediated Research [25].

### 2.3. Inclusion/Exclusion Criteria

The following inclusion criteria were applied: being a heterosexual woman in the third trimester of a healthy pregnancy, being at least 18 years of age at the time of their admission to the study, having a lack of past or current clinical diagnoses of any psychiatric disease including depression, being someone who is planning natural birth in a hospital, having no past traumatic childbirth experience, there being no COVID-19 infection among the respondents or their immediate family at the time of their admission, having no quarantine experience during their participation in the study, having a stable relationship, and giving their signed, informed consent for participation in the study. Women were excluded if their pregnancy was high risk or if they were below 36 weeks of gestation when they were younger than 18 years old, if they were single mothers, planning cesarean section, had a past traumatic childbirth experience, were suffering from any mental or somatic diseases including coronavirus, or if they were being quarantined. The criteria for the absence of COVID-19 among their closest relatives and the lack of quarantine experience during the study were introduced to eliminate those factors that could additionally affect the women’s mental health. The study was designed to include only expectant mothers who underwent similar experiences of the lockdown during the COVID-19 pandemic.

### 2.4. Procedure and Data Collection

The current data were collected via an online survey from June to September 2020. The recruitment began with the distribution of leaflets describing the study and inviting women to participate in antenatal schools and gynecology clinics. Any women who were interested in participating were asked to send an e-mail to the address that was provided by the leaflet. Those who were interested signed an electronic informed consent form to receive a personalized link to the survey. Initially, a total of 132 expectant mothers were interested in participating in the study: of these, 39 were rejected at the recruitment stage due to their failure to meet the inclusion criteria (i.e., pregnancy ≥ 36 weeks gestation, high-risk pregnancy, younger than 19 years old, or single motherhood). Of the remaining 93 volunteers, 13 returned incomplete questionnaires (they only entered the sociodemographic and/or gynecological–obstetrics data). Finally, 80 expectant mothers who met the eligibility criteria were included in the analyses.

### 2.5. Measures

The sociodemographic and medical/obstetric factors survey included questions on their maternal age, level of education, financial status, relationship status, personal health history, gestational week, course of pregnancy (healthy vs. high-risk), the mother’s health during pregnancy, or whether the current pregnancy was planned.

The COVID-19 exposure and pandemic impacts survey collected information about their personal and family experience of COVID-19, current physical distancing/isolation situation including current work status, and income loss. Table 1 presents the details of the collected data.

The Pandemic-Related Pregnancy Stress Scale (PREPS) by Preis et al. [15] is a novel instrument for assessing the thoughts and concerns that pregnant women might have due to the COVID-19 pandemic and its related impacts. It contains 15 items that are rated on a scale from 1 = Very Little to 5 = Very Much. The tool has a three-factor structure, comprising Perinatal Infection Stress (I-S), Preparedness Stress (P-S), and Positive Appraisal. It has demonstrated acceptable to good reliability (α’s from 0.68 to 0.86). The sum of the first (I-S) and second (P-S) scales represents the total pandemic-related pregnancy stress scores (total PREPS). The Polish version of the PREPS has also showed that it has good psychometric properties: the reliability, which was measured with Cronbach’s alpha, was 0.86 for the total score, with subscale values that were ranging from 0.69 to 0.88 [26]. The Cronbach’s alpha coefficients for the current study were 0.91 for total scale reliability and from 0.68 to 91 for the subscales.

The Edinburgh Postnatal Depression Scale (EPDS) by Cox, Holden, and Sagovsky [27] is a self-reporting questionnaire that is widely used among pregnant and postpartum women to assess the feelings that they have experienced during the past seven days. The EPDS contains 10 items, and each are scored from 0 to 3. The answers reflect the woman’s degree of agreement with the statements, and the total score ranges from 0 to 30: a higher score indicates greater severity of the perinatal depressive symptoms. Cox et al. [27] recommended using a cut-off value of 12/13; however, a recent meta-analysis by Levis et al. [28] found that an EPDS cut-off value of 11 or a higher value for the maximized combined sensitivity and specificity. They recommended that if the intention is to avoid the occurrence of false negatives and to capture all of the participants who might meet the diagnostic criteria based on a further evaluation, then this lower cut-off value might be preferred. Therefore, a cut-off value of 11 was used in our study. In the current sample, Cronbach’s alpha reliability coefficient was satisfactory, and it amounted to 0.89.

The Traumatic Birth Perception Scale (TBPS) was developed by Yalniz et al. [29]. The TPBS is a self-reporting questionnaire that is aimed to determine the traumatic birth perception of the respondents. It contains 13 items (questions) concerning the physical, emotional, and mental trauma of childbirth. Every question is scaled from zero (0) to ten (10), corresponding to a non-existent to the most severe fear/concern. The minimum and maximum achievable scores on this scale are 0 and 130 points, respectively, and the mean total score of the scale represents the level of traumatic birth perception. The study of this questionnaire’s development [29] found that a mean total score of 0–26 indicates that there is a very low level of traumatic birth perception, 27–52 indicates that there is a low level, 53–78 indicates that there is a moderate level, 79–104 indicates that there is a high level, and 105–130 indicates that there is a very high level. The Cronbach’s alpha reliability coefficient of the scale was 0.89 in the original study. A preliminary validation of the TBPS among 778 Polish women found that the scale had a single-factorial structure and satisfactory internal consistency (α = 0.90). The Cronbach’s alpha in the current study was 0.91.

The Connor-Davidson Resilience Scale (CD-RISC-10) by Connor and Davison [30] is a measure of resilience, which is understood to be a stress coping ability, and it comprises five components: Personal Competence, Acceptance of Change and Secure Relationships, Trust of One’s Instincts, Tolerance of Negative Effect, Strengthening Effects of Stress, Control, and Spiritual Influences. However, the short version that was used in the study is univariate [30], with 10 items that were rated on a scale of 0 (completely false) to 4 (completely true). The sum of all of the responses ranges from 0–40, with 40 indicating the highest level of resilience. In the shortened version of the CD-RISC, the Cronbach’s alpha reliability coefficient is 0.85 [31]. In the studied sample, the Cronbach’s alpha reliability coefficient was satisfactory, and it amounted to 0.93.

### 2.6. Data Analysis

Statistical analyses were conducted using SPSS 27 and PROCESS macro version 4.0 for SPSS. The demographic characteristics were summarized as the mean (M) with standard deviation, (SD) for continuous variables, and as frequency counts (percentages) for the categorical variables. As the participants were divided into two groups according to their resilience level, a chi-square test was used to evaluate the frequency differences between their demographic variables. Regarding the distribution of the variables, only the Preparedness stress and Positive appraisal were not normally distributed. However, the skewness and kurtosis were also analyzed and none of the coefficient values exceeded the value +/−1. Therefore, parametric tests were used: an independent *t*-test was used to calculate the differences between the means, and a Pearson’s correlation coefficient was used to establish the links between the variables. In the case of the comparisons in the subgroups that were unequal and/or did not meet the assumptions of homogeneity of variance, non-parametric Mann-Whitney and Kruskal–Wallis tests were used. The Cohen’s d was used to determine the effect size for two means.

The mediation analysis was performed using the PROCESS macro by Hayes [32]. The mediating role of resilience (mediator) was tested in the relationship between the traumatic childbirth perception and pandemic-related stress (predictors; independent variables) and prenatal depression symptoms (dependent variable). The procedure that was used was the bootstrapping method that was proposed by Hayes [32], which involved drawing 5000 bootstrap samples. The mediation procedure provides a more comprehensive view of a complex structure in which an independent variable, that is acting as a predictor, is linked to a dependent variable via a third variable, that is acting as a mediator. Mediating effects occur when the mediating variable decreases the predictive power of the independent variable for the dependent variable. In the current analysis, pandemic-related stress and a traumatic childbirth perception were used as predictors. Resilience acted as a mediator, and the prenatal depression symptoms were the dependent variable. An a priori power analysis using G*Power indicated that a total sample size of *n* = 55 was required to provide 80% statistical power with α = 0.05 and detect a medium (f2 = 0.15) effect size for two predictors. The statistical significance level was set at *p* < 0.05.

## 3. Results

### 3.1. Study Sample Characteristic

The study group comprised 80 pregnant women who were aged from 21 to 39 years old (M = 28.6; SD = 3.2). The mean age of gestation was 32.9 (SD = 4.3). The majority of the sample (76.3%) completed a higher education course, lived in a large city (65%), were in a marital relationship (76.3%), were without other children (38.8%), and did not report previous fertility problems (77.5%). Most of the current pregnancies were planned (83.8%) and were without any complications (76.3%). A total of 48 women (60%) assessed their economic situation as good and unchanged, despite the COVID-19-related lockdown. The results of the comparisons between the subgroups due to the sociodemographic variables (Mann-Whitney U tests or Kruskal–Wallis tests) showed that there were no differences in the scope of the prenatal depression scores. Table 1 presents the detailed characteristics of the studied sample, concerning the outcome variable (the prenatal depression scores).

### 3.2. Resilience Level in the Study Sample

The overall resilience scores were lower when they were compared to that of the normative data from an American study, where the median score was 32, and the lowest to highest quartiles were 0–29, 30–32, 33–36, and 37–40, respectively [31]. The median value of the general level of resilience in our study sample was 20.0 (M = 19.5, SD = 8.6). We defined the quartiles as four groups of equal numbers that were taken from the observed distribution of the scores. Taking into account that the first quartile (Q1) describes the score range for the lowest group (lowest 25% of the population), i.e., the least resilient, the second (Q2) and third (Q3) describe the intermediate scores, and the fourth (Q4) describes the highest or most resilient women (above 75% of the population); the following quartiles ranges that were obtained are listed from the lowest to the highest: 0–14, 15–20, 21–27, and 28–36, respectively. The number of the groups that were formed as a result of their division into quartiles was very similar. In total, 23.8% of women (*n* = 19) were the least resilient, 27.5% represented the lower intermediate values for resilience (*n* = 22), and 25% had a higher intermediate resiliency (*n* = 19), and 23.8% were the most resilient (*n* = 19). To allow for further comparative analyses, two groups were created: the lower resilience group (*n* = 41) which included women from the Q1 and Q2 groups, and a higher resilience group (*n* = 39) whose results included the women from the Q3 and Q4 groups. The chi-square analysis showed no differences in the scope of the analyzed variables between the groups. Table 2 presents the detailed sociodemographic characteristics of the studied sample, including the women with the higher and lower levels of resilience.

### 3.3. Prenatal Depression Symptoms in a Study Sample

The mean depression score obtained from all participants was 9.43 (SD = 6.12). McCabe-Beane et al. [33] recommend the following ranges for EPDS scores interpretation: 0 to six points indicate none or minimal depression, seven to 13 indicate mild depression, 14 to 19 indicate moderate, and 20 to 30 severe depression. Levis et al. [28] recommend a cut-off value of 11 for EPDS scores indicating the presence of depression symptoms. Based on the first criterion, 33 pregnant (41.3%) women demonstrate a mild risk of depression symptoms, 13 (16.3%) moderate, and seven (8.8%) severe. Based on the second Levis et al. [28] criterion, 39 subjects (48.8%) demonstrated a higher risk of depression. Significantly higher mean depression symptom scores (t (78) = 4.668; *p* < 0.001) were observed in women with a low level of resilience (M = 12.2; SD = 6.1) compared to those with higher resilience (M = 6.5; SD = 4.7); the strength of the effect was very high (Cohen’s d = 1.05).

### 3.4. Pandemic-Related Stress in a Study Sample

The mean pandemic-related pregnancy stress total scores obtained by the examined women was 2.81 (SD = 1.10), which, assuming a range of 1 to 5 points, can be considered as being in the middle of the scale. The analyses revealed partial differences in the experience of prenatal stress associated with the COVID-19 pandemic by women depending on the level of their resilience. A statistically significant difference was found in the Preparedness stress and total PREPS level (see: Table 3). Women from the lower resilience group showed a higher intensity of total stress (t(78) = 2.007, *p* < 0.05; d = 0.56), and preparedness stress (t(78) = 2.448, *p* < 0.05; d = 0.45); however, the effect size was moderate in both cases. There were no statistically significant differences between the groups in terms of infection stress and positive appraisal.

### 3.5. Traumatic Childbirth Perception in a Study Sample

The mean TBPS score (see: Table 4) that was obtained from the participants in the current study was 57.30 (SD = 26.21), and this does not differ significantly from the data that were obtained in a recent study by Türkmen et al. [4], in which the mean score was 63.52 (SD = 27.12) for pregnant women who were expecting a vaginal birth, and 65.65 (SD = 25.64) for pregnant women who were expecting a Caesarean birth. Based on the suggested cut-off point for the TBPS, the traumatic childbirth perception levels were very low in 12.5% of the participants, low in 33.7% of them, moderate in 31.3% of them, high in 18.7% of them, and very high in 3.8% of them. Traumatic birth perception was significantly higher (t (78) = 6.036; *p* < 0.001) in women with a low level of resilience (M = 71.6; SD = 24.2) when they were compared to those with a higher resilience (M = 42.2; SD = 18.9). The Cohen’s d was 1.35, indicating that there was a very high effect size.

### 3.6. Relationship between the Traumatic Childbirth Perception, Pandemic-Related Pregnancy Stress, and Prenatal Depression Symptoms

After assessing the descriptive statistics of all of the analyzed variables (Table 4), a Pearson’s correlation analysis was performed to determine the relationships between the variables (Table 5). The strongest relationship was found between prenatal depression symptoms and resiliency. The relationship was negative, indicating that a higher level of resiliency is accompanied by a lower level of prenatal depression symptoms (r = −0.62; *p* < 0.001). A positive link was identified between the traumatic childbirth perception and prenatal depression symptoms (r = 0.51; *p* < 0.001), indicating that the level of depression symptoms increases with the level of traumatic childbirth perception. Of all of the pandemic-related pregnancy stress factors, the highest correlation coefficient that was obtained with prenatal depression symptoms was for the Preparedness stress (r = 0.38; *p* < 0.001). The abovementioned relationships were found to be of high or moderate strength.

### 3.7. Resilience as a Mediator between Pandemic-Related Stress/Traumatic Childbirth Perception and Prenatal Depression Symptoms

To explore the more complex relationship between prenatal depression, traumatic childbirth perception, pandemic-related stress, and resilience, a mediation analysis was performed with the EPDS total scores as the dependent variable (DV) and resilience as the mediator. Before testing the mediating effect of resilience in the relationship between pandemic-related stress and prenatal depression, and between traumatic childbirth perception and prenatal depression, a regression analysis was performed to determine the direct effect of resilience, traumatic childbirth perception, and pandemic-related stress on prenatal depression symptoms.

The analysis revealed a significant direct effect between pandemic-related stress and prenatal depression (βc = 0.285, SE = 0.05, t = 2.63, *p* < 0.05) as well as between pandemic-related stress and resilience (βa = −0.283, SE = 0.07, t = −2.61, *p* < 0.05) and between resilience and prenatal depression (βb = −0.585, SE = 0.07, t = −6.34, *p* < 0.001) (see: Figure 1). After the introduction of resilience as a mediator, the strength of the relationship not only decreased but also it ceased to be statistically significant (βc′ = 0.120, SE = 0.04, t = 1.29, *p* = 0.19), which indicates that it is in a state of full mediation (R^2^ = 0.39, F = 25.31, *p* < 0.001; Z = 2.43, *p* < 0.05). The results indicate that in pregnant women, a high level of resilience protects them from the effects of pandemic-related stress on perinatal depression symptoms.

Similarly, we found a significant direct effect between traumatic childbirth perception and prenatal depression (βc = 0.512, SE = 0.02, t = 5.26, *p* < 0.001) as well as between traumatic childbirth perception and resilience (βa = −0.623, SE = 0.03, t = 7.03, *p* < 0.001) and between resilience and prenatal depression (βb = −0.491, SE = 0.08, t = −4.39, *p* < 0.001) (see: Figure 2). After the introduction of resilience as a mediator, the strength of the relationship between traumatic childbirth perception and prenatal depression symptoms ceased to be statistically significant (βc′ = 0.206, SE = 0.043, t = 1.84, *p* = 0.07), which indicates that it is in a state of full mediation (R2 = 0.41, F = 26.69, *p* < 0.001; Z = 3.71, *p* < 0.001). The results indicate that in pregnant women, a high level of resilience protects them from the effects of traumatic childbirth perception on perinatal depression symptoms.

## 4. Discussion

The outbreak of the COVID-19 pandemic brought with it an accumulation of stressful events, concerns for health and life, and isolation. Further difficulties were also associated with social distancing, limited access to medical care, and uncertainty regarding the future [13,34,35]. In addition, pregnant women were faced with limitations that were related to the involvement of family in the delivery of the child (the child’s father cannot participate in the delivery), and the possible separation of the child from the mother, without the possibility of breastfeeding in the event of one of them contracting the virus [36,37,38]. Pregnant women were even more acutely exposed to problems that were related to mental health hazards, such as stress, anxiety, and depression [26,39], and they required special protection against COVID-19 [40]. Previous studies have shown that such serious threats as natural disasters, armed conflicts, or sudden and unforeseen situations increase the frequency of mental health problems among pregnant women [41,42].

Thus, the present study was based on three assumptions. The first concerns the levels of prenatal depression, pandemic-related pregnancy stress, and traumatic childbirth perception in low- and high-resilience pregnant women. The second assumption concerned the nature of the relationship between the analyzed variables. Finally, the latter assumption was that resilience had a mediating effect on the relationship between prenatal depression and its two risk factors: pandemic-related pregnancy stress and traumatic birth perception

### 4.1. Prenatal Depression

Overall, nearly 50% of the women in our study demonstrated symptoms of prenatal depression, with the mean value in the low-resilience group being almost twice as high as it was in the high resilience group. Other studies confirm that women suffer from depression, in general [43,44], in the postpartum period [45,46], and antenatally, as well [47,48], they have lower resilience levels. It is worth paying attention to the occurrence of prenatal depression symptoms depending on sociodemographic factors. Although there were no statistically significant differences in any of the sociodemographic variables, the level of the depressive symptoms was higher in the group of women with a healthy, uneventful pregnancy when it was compared to that of the high-risk pregnancies. This result is in contradiction with previous reports [49]. This is probably due to the discrepancies in the size of the two groups (61 healthy vs. 19 high-risk). However, it is also a hint to especially control this variable in subsequent investigations.

### 4.2. Traumatic Childbirth Perception

At least a moderate level of traumatic childbirth perception was present in over 55% of the respondents. This percentage is much higher than that which was reported in other studies involving the populations of different nationalities. For example, 11.1% of a total of 475 pregnant women in Eastern Sudan were found to experience a severe fear of childbirth [50], while a severe fear of childbirth was noted in 5.3% of women in a study in Ireland, with a high fear level of 36.7% [51]. The systematic review found that national rates of the fear of childbirth to vary from 6.3 to 14.8% in nine European countries, Australia, Canada, and the United States [52]. However, it should be noted, that in all of the above-mentioned studies, the most frequently-used scale for the fear of childbirth was the Wijma Delivery Expectancy Questionnaire [53], thus, it is not possible to compare them directly with our present findings.

The TBPS, that measures the traumatic perception of childbirth, is a novel instrument, and as it was developed originally on a Turkish population, only a few studies are available for a comparison with ours [4,29,54]. The TBPS also has not been used in Polish studies so far; the most frequently-used tool is the Birth Anxiety Questionnaire (Kwestionariusz Lęku Porodowego; KLP-II) by Putyński and Paciorek [55]. Based on the KLP-II, Kaźmierczak et al. [56] reported that 65.66% of studied pregnant women were characterized by a low/average level of labor anxiety, 18.18% had an increased level of it, 10.10% a high level of it, and 6.06% a very high level of it. Although these results seem similar to those that were obtained in our present study, the key to understanding our results is the specific nature of the chosen tool. TBPS items relate not only to concerns that are related to the course of labor or recovery after delivery (as is the case with KLP-II), but they also address the somatic symptoms that are related to thinking and imagining the birth, in general, and not just the respondent’s experience. The perception of traumatic childbirth was significantly higher among the low resilience group in our study. This result seems understandable, as resilience in the perinatal period is defined as the ability to protect oneself against negative thoughts, minimize the impact of fear or anxiety, and promote recovery from stressor events [57]. In a recent study, Huang et al. [58] found a similar relationship between resilience and fear of childbirth; a lower resilience is associated with a stronger fear of childbirth among the examined pregnant women in China.

### 4.3. Pandemic-Related Pregnancy Stress

The values of the scores in our study were slightly lower than the data that were obtained by a validation study for the Polish adaptation of PREPS [26]. Similarly to the Spanish [59] and German studies [60], the highest score among the three dimensions of PREPS was recorded for the Preparedness stress. Our findings indicate that pregnant women with a high resilience were more likely to have a lower general level of pandemic-related pregnancy stress, and prenatal stress that was related to labor preparation, planning, and care than those with a low resilience did. Interestingly, the women did not differ in terms of the prenatal stress that they experienced in relation to infection. These results seem to be consistent with those regarding the traumatic perception of childbirth; they suggest that during the COVID-19 lockdown, pregnant women were more worried about the course of labor than the risk of the coronavirus disease. Due to the restrictions that were related to the transmission of the coronavirus, some countries, including Poland, introduced restrictions on accompanying the mother during childbirth. This situation raised concerns among women as to the course of the delivery, i.e., many of them reported being worried about not feeling safe in the hands of the medical staff without the presence of the child’s father, as was planned. An additional factor that was intensifying the stress was the fear that if the newborn had to remain in the hospital after delivery for medical reasons, the mother would not be able to accompany it. Based on the definition of resilience that is described above, a high level of it would probably enable the mother to better cope with stress that was aggravated by a pandemic situation. Indeed, Dikmen-Yildiz et al. [61] reported that resilient women reported more satisfaction with healthcare professionals, and less depression and fear of childbirth.

### 4.4. Relationship between the Resilience and Prenatal Depression Symptoms, Traumatic Childbirth Perception, and Pandemic-Related Pregnancy Stress

Many studies have confirmed that resilience is associated with stress, including traumatic stress, such as those of traumatic childbirth experiences [62]. A higher level of resilience also accompanies good mental health, especially so in the perinatal period. Conversely, a low resilience, which is linked to various risk factors such as stress and/or severe fear of childbirth, may be associated with prenatal depression [47,48]. In our study, a negative correlation between resilience and the other variables was observed, which can be also caused by pandemic situations.

Our latter assumption was that resilience had a mediating effect on the relationship between prenatal depression and its two risk factors: pandemic-related pregnancy stress and traumatic birth perception. It has previously been found that resilience protects from continuous stress and reduces the risk of mental illnesses during COVID-19 pandemics [63], and acts as a mediator between pandemic-related stress and depression and anxiety symptoms [64]. The results of our study demonstrate that the relationships between pandemic-related stress with traumatic childbirth perception and with prenatal depression were fully mediated by resilience during pregnancy.

Existing studies on the stress that is experienced by pregnant women during epidemics, floods, earthquakes, or other natural disasters covered a short period and a limited territory [65]. In this regard, the COVID-19 pandemic is an exceptional situation, as for many months, the situation in the world has been very uncertain, with many unknowns in various life spheres. Concerns about the uncertainties surrounding many life areas arise from the stress of childbirth, which in pandemic conditions, may be exacerbated by the limited possibilities of birth with the child’s father being present, limited contact with the child after childbirth, and the lack of constancy between hospitals. In those circumstances, pregnant women, as a vulnerable group, are more exposed to depression, anxiety, and stress but also there is an increased risk of negative consequences for their infants, such as a less secure attachment with the infant. Our findings indicate that during those difficult circumstances, resilience protects pregnant women from pandemic stress, traumatic childbirth perception, and prenatal depression, which also often lead to postpartum depression. In consequence, resilience helps mothers to build a bond with their infant and support a secure attachment when she is attentive to signals from their infant. A previous study reported that the presence of maternal depressive symptoms during pregnancy is a strong risk factor for postpartum depression [66]. Hence, there is a strong need to implement social programs that enhance resilience among new parents, especially in the social sphere to improve social support [67]. Indeed, building resilience should be an important element of the mental health promotion interventions that are targeted at women who are expecting a baby [68], especially in such a crisis as that of a coronavirus pandemic.

### 4.5. Strenghts and Limitations

The strengths of our research include the use of well-validated measurement tools. First, we used an internationally validated depression screening instrument (EPDS), which has a good diagnostic performance which is confirmed by other studies. Second, we used a scale to measure the stress that is typically associated with experiencing pregnancy during the coronavirus pandemic (PREPS). This allowed us to examine the level of stress that was associated with the specificity of this new, difficult, and unexpected situation, not just the overall level of stress that may have other roots. Third, we used a new tool to measure the traumatic childbirth perception (TPBS), which assesses the level of perceiving childbirth as traumatic not only in women who have given birth and may have negative memories of the course of childbirth. The TPBS allows for the exploration of general attitudes and perceptions about childbirth. We assumed that it is important, especially in the period of a pandemic, when restrictions that are related to the course of childbirth have been introduced, especially regarding the presence of a partner during the labor. Finally, we assessed not only the differences in the severity of prenatal depressive symptoms depending on them having low and high resilience, but also, a more complex relationship: whether resilience could change (reduce or eliminate) the relationship between stress and a traumatic perception of childbirth and depression.

However, although the present study offers important new information regarding the risk and protective factors of prenatal depression symptoms among pregnant women, it has some limitations which should be noted. First, the cross-sectional design of the study did not indicate the temporal and causal relationships between the variables, which also limits the interpretation of the findings. Additionally, the lack of a follow-up study makes it impossible to see if postnatal depression is also correlated with the results. Second, the study sample is relatively small. This is most likely because the current data were collected from June to September 2020, at the turn of the first and second waves of the coronavirus pandemic in Poland: a time of adjusting to functioning in a remote reality. As such, the participation in research was probably not a priority, especially during pregnancy. With such a small number of respondents, it is difficult to achieve a high diversity, and the sample may not be representative of the total population. The majority of participants were well-educated, married women with a good financial status, living in large urban areas, which could call into question the generalizability of the findings. In further explorations, it is necessary to diversify the study participants in terms of the sociodemographic factors, including a lower SES, to obtain more generalized results. Another possible limitation is that we only recruited women who were willing and able to submit an email to participate in an online survey. This could pose a problem of selection bias. Internet research makes it impossible to obtain information from participants who, for various reasons, do not have access to the internet, are not users of social media, or are unable to use information technology fluently [69]. Another aspect to consider is that the study had a single measurement time point that was very early in the pandemic. There is, therefore, no investigation in the following waves, which were faced by different policies, that could affect the pregnancy experience during the pandemic. Also, only self-reporting measures were used to evaluate the prenatal depression symptoms. Structured diagnostic interviews should also be included in further studies. The final limitation is that we tested only resilience as a mediating factor between pandemic-related pregnancy stress or traumatic childbirth perception and prenatal depression symptoms. It must be assumed, however, that there are other important factors affecting the relationship between these variables. For example, we did not control for the social and medical support that they received, which can significantly change the nature of the analyzed relationships, and should be which should be included in further explorations.

## 5. Conclusions

This cross-sectional study was conducted among women in the third trimester of pregnancy to examine the potential mediating role of resilience on the influence of prenatal depression on traumatic childbirth perception and pandemic-related stress during pregnancy. Our findings demonstrate that the associations between the abovementioned variables and prenatal depression were fully mediated by resilience. These data underscore the importance of mental health interventions that enhance the resilience among pregnant women, especially those who experience high levels of pandemic-related stress and/or for whom childbirth is a traumatic event, to decrease their risk of prenatal depression.

## Figures and Tables

**Figure 1 behavsci-12-00371-f001:**
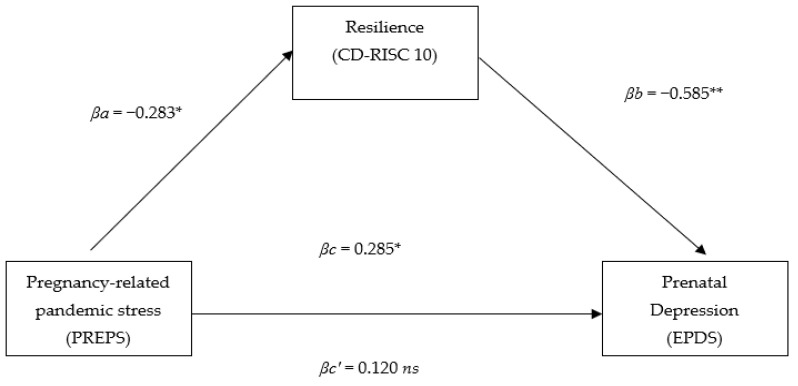
Model of relationships between pandemic-related stress, resilience, and prenatal depression in the study sample. * *p* < 0.05, ** *p* < 0.001, *ns*—non-significant.

**Figure 2 behavsci-12-00371-f002:**
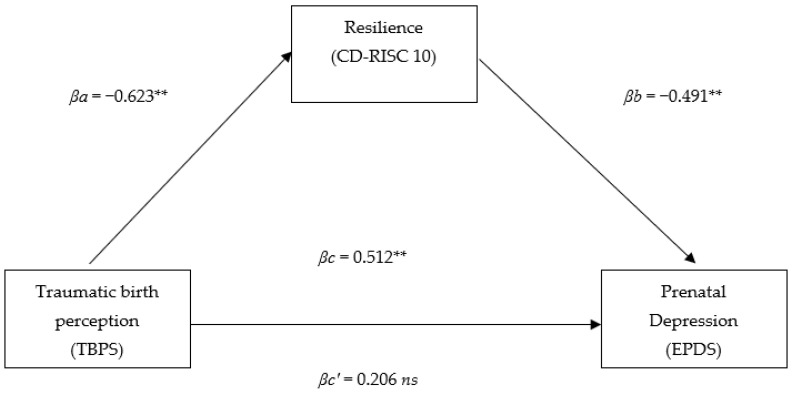
Model of relationships between traumatic birth perception, resilience, and prenatal depression in the study sample. ** *p* < 0.001; *ns*—non-significant.

**Table 1 behavsci-12-00371-t001:** Study sample characteristics with regard to prenatal depression scores (*N* = 80).

Variable		*n* (%)	EPDS Scores				
			*M*	*SD*	Mean Rank	*U*/*H*	*p*
Education						551.000	0.747
	High school and lower	19 (23.8)	8.6	5.1	39.0		
	University degree	61 (76.3)	9.7	6.4	41.0		
Place of residence						2.118	0.347
	Rural	10 (12.5)	10.5	6.5	44.5		
	Small city	18 (22.5)	7.4	5.4	33.6		
	Large city	52 (65.0)	9.9	6.2	42.1		
Marital status						570.500	0.919
	Marital	61 (76.3)	9.4	5.9	40.7		
	Partnership	19 (23.8)	9.5	6.7	40.0		
Children						1.509	0.470
	None	31 (38.9)	9.1	6.3	39.0		
	One	26 (32.5)	8.7	5.9	37.9		
	Two	23 (28.7)	10.7	6.1	45.5		
Economic situation						0.695	0.706
	Very good	14 (17.5)	10.7	7.9	44.1		
	Good	48 (60.0)	9.4	5.7	49.7		
	Average	18 (22.5)	8.4	5.9	37.3		
Current pregnancy status						419.000	0.069
	High risk	19 (23.8)	7.5	6.8	32.1		
	Health	61 (76.3)	10.0	5.8	43.1		
Previous fertility problems						458.500	0.251
	Yes	18 (22.5)	8.4	7.3	35.0		
	No	62 (77.5)	9.7	5.8	42.1		
Planned pregnancy						300.000	0.077
	Yes	67 (83.8)	9.0	6.4	38.5		
	No	13 (16.3)	11.4	4.3	50.9		
COVID-19 impact on the family’s economic situation							
	Improved	3 (3.8)	7.7	4.0	34.5	2.827	0.243
	Not changed	56 (70.0)	8.8	6.2	38.1		
	Worsened	21 (26.3)	11.3	6.1	47.7		

*U*—Mann Whitney test result; *H*—Kruskal–Wallis test result.

**Table 2 behavsci-12-00371-t002:** Sociodemographic characteristics concerning lower and higher resilience groups.

Variable		Lower Resilience *n* = 41		Higher Resilience *n* = 39		*Χ* ^2^	*df*	*p*
		*n*	%	*n*	%			
Education						2.070	1	0.150
	High school and lower	7	17.1	12	30.8			
	University degree	34	82.9	27	69.2			
Place of residence						3.044	2	0.218
	Rural	6	14.6	4	10.3			
	Small city	6	14.6	12	30.8			
	Large city	29	70.7	23	59.0			
Marital status						0.150	1	0.698
	Marital	32	78.0	29	74.4			
	Partnership	9	22.0	10	25.6			
Children						0.438	2	0.803
	None	17	41.5	14	35.9			
	One	12	29.3	14	35.9			
	Two	12	29.3	11	28.2			
Economic situation						2.066	2	0.356
	Very good	9	22.0	5	12.8			
	Good	25	61.0	23	59.0			
	Average	7	17.1	11	28.2			
Current pregnancy status						0.150	1	0.698
	High risk	9	22.0	10	25.6			
	Health	32	78.0	29	74.4			
Previous fertility problems						0.015	1	0.904
	Yes	9	22.0	9	23.1			
	No	32	78.0	30	76.9			
Planned pregnancy						1.016	1	0.313
	Yes	36	87.8	31	79.5			
	No	5	12.2	8	20.5			
COVID-19 impact on the family’s economic situation						0.998	2	0.607
	Improved	2	4.9	1	2.6			
	Not changed	30	73.2	26	66.7			
	Worsened	9	22.0	12	30.8			

**Table 3 behavsci-12-00371-t003:** Comparison of the prenatal stress related to the COVID-19 pandemic measured by PREPS in groups with low and high levels of resilience.

Variable	Lower Resilience		Higher Resilience		*t*	*p*	*Cohen’s d*
	*M*	SD	*M*	SD			
Infection stress	12.0	5.3	10.8	6.1	0.986	0.327	NA
Preparedness stress	24.6	8.1	19.9	8.6	2.448	0.017 *	0.56
Positive appraisal	6.3	2.6	7.2	3.5	−1.179	0.242	NA
Pandemic-related pregnancy stress (total PREPS)	36.6	12.1	30.7	13.9	2.007	0.048 *	0.45

* *p* < 0.05; NA—not applicable.

**Table 4 behavsci-12-00371-t004:** Descriptive statistics of analyzed variables (*N* = 80).

Variables	*M*	*SD*	Range of Scores	*LL*; *UL* *95% CI*	Skewness	Kurtosis
Prenatal depression (EPDS)	9.43	6.12	0–23	8.06; 10.79	0.46	0.53
Resilience (CD-RISK)	19.95	8.60	0–36	18.04; 21.86	−0.23	−0.65
Perinatal Infection Stress (PREPS)	2.28	1.14	1–5	2.03; 2.54	0.16	−0.74
Preparedness Stress (PREPS)	3.19	1.23	1–5	2.91; 2.46	−0.47	−0.78
Positive Appraisal (PREPS)	2.25	1.03	1–5	2.02; 2.47	0.32	−0.53
Pandemic-related stress (total PREPS)	2.81	1.10	1–5	2.57; 3.06	−0.38	−0.76
Traumatic childbirth perception (TPBS)	57.30	26.21	11–112	51.47; 63.13	0.27	−0.76

*M*—mean; *SD*—standard deviation; *95% CI* = Confidence Intervals; *LL*—lower level; *UL*—upper level.

**Table 5 behavsci-12-00371-t005:** Correlation matrix for the variables in the mediation model test.

Variable	1	2	3	4	5	6
1. Prenatal depression (EPDS)	1					
2. Resilience (CD-RISK)	−0.62 **	1				
3. Perinatal Infection Stress (PREPS)	0.09	−0.12	1			
4. Preparedness Stress (PREPS)	0.38 **	−0.36 **	0.70 **	1		
5. Positive Appraisal (PREPS)	−0.18	0.10	0.21	0.23 *	1	
6. Pregnancy-related stress (total PREPS)	0.29 *	−0.28 *	0.89 **	0.95 **	0.24	1
7. Traumatic birth perception (TBPS)	0.51 **	−0.62 **	0.14	0.42 **	0.07	0.34 **

* *p* < 0.05, ** *p* < 0.001.

## Data Availability

The datasets used and analyzed during the current study are available in an OSF data repository at https://osf.io/uayf2/?view_only=534346c077984ac5af41f2fd3732f7f5 (accessed on 25 August 2022).

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
