# Peer review of "Experiencing Pregnancy during the COVID-19 Lockdown in Poland: A Cross-Sectional Study of the Mediating Effect of Resiliency on Prenatal Depression Symptoms"

_behavsci, 2022, doi:10.3390/bs12100371_

Round 1
Reviewer 1 Report
This study has aimed at investigating the protective effect of relationship resilience on prenatal depression during the first wave of COVID-19 in Poland. The study was conducted after the first wave. The article seems interesting; however, I have some considerations:
Major considerations:
1. Was the sample size calculated? Please indicate this in the methods.
2. Please indicate the novelty limitation of this manuscript in the limitations section. It appears to be done very early in the pandemic. There was also no investigation in the following waves, which were faced by different policies. Finally, there was no follow-up to see if postnatal depression is also correlated.
Minor considerations:
1. I couldn't understand the aim in the abstract. It is poorly formulated. Please re-write it.
2. What is relationship resilience? A definition and an explanation of its measurement should be included in the abstract as well. I couldn't understand the abstract until I read the full text. This should not be the case.
3. The abstract has not mentioned which statistical tests were done. The authors just simply barged into the results. Some basis for interpretation should be provided to the reader.
4. There are various proofreading errors. I suggest that the authors ask a native speaker to edit and proofread the document. Or they can use some free proofreading software like Grammarly. For example, in the very first sentence in the abstract, the authors said: "uncertainty for" but it should be "uncertainty to".
Author Response
This study has aimed at investigating the protective effect of relationship resilience on prenatal depression during the first wave of COVID-19 in Poland. The study was conducted after the first wave. The article seems interesting; however, I have some considerations:
ANSWER: We appreciate the time you spend preparing a review. Thank you for all the valuable comments that increase the scientific level of the article.
All changes made to the text are marked in blue
Major considerations:
- Was the sample size calculated? Please indicate this in the methods.
ANSWER: Following sample size information was included in the manuscript:
An a priori power analysis using G*Power indicated that a total sample size of n = 55 was required to provide 80% statistical power with α=.05 and detect a medium (f2=.15) effect size for two predictors. The power analysis for the mediation model was applied.
- Please indicate the novelty limitation of this manuscript in the limitations section. It appears to be done very early in the pandemic. There was also no investigation in the following waves, which were faced by different policies. Finally, there was no follow-up to see if postnatal depression is also correlated.
ANSWER: As recommended, the information was added to the Limitations section.
Minor considerations:
- I couldn't understand the aim in the abstract. It is poorly formulated. Please re-write it.
ANSWER: We agree with this comment, thank you. The aim was reformulated.
- What is relationship resilience? A definition and an explanation of its measurement should be included in the abstract as well. I couldn't understand the abstract until I read the full text. This should not be the case.
ANSWER: Thank you for this valuable comment. The abstract has been supplemented with an explanation of what resilience is. We have not included information about resilience measurement in the Abstract so that the number of characters does not exceed the required threshold. However, we hope that after the introduction of the explanation of what resilience is, the content of the Abstract will be clear enough.
- The abstract has not mentioned which statistical tests were done. The authors just simply barged into the results. Some basis for interpretation should be provided to the reader.
ANSWER: Thank you. The information was included.
- There are various proofreading errors. I suggest that the authors ask a native speaker to edit and proofread the document. Or they can use some free proofreading software like Grammarly. For example, in the very first sentence in the abstract, the authors said: "uncertainty for", but it should be "uncertainty to".
ANSWER: The manuscript was previously checked by an English native speaker, however, following the recommendation, we used Grammarly software to make corrections.
Reviewer 2 Report
Thank you for giving me the opportunity to review this manuscript.
This cross-sectional study describes potential impact of resilience on depression in Polish women with pregnancy during the COVID-19 pandemic.
I think it is necessary to revise the manuscript.
-major
1) Please describe the study design by using the PECO format. The Population is the Polish participants with pregnancy. The exposure is low resiliency. The control is high resiliency. The primary outcome is depressive symptoms. Please delete the sentences that the resiliency is the mediator.
2) Please describe what is potential confounders, and any effect modifiers. It is impossible to assess how the potential confounders affect the outcomes. For example, please describe unadjusted estimates and their precision (Model 1), those after adjusting for baseline characteristics (Model 2), and those after adjusting for baseline characteristics + potential confounders (stress scales and so on) (Model 3). Please make clear what kind of confounders was adjusted for and why they were included. Actually, table 4 showed that depressive scores were correlated with pandemic-related pregnancy stress scales and traumatic child blrth perception. I think that this result indicated that with pandemic-related pregnancy stress and traumatic child birth perception would be potential confounders of the association between resilience and depressive symptoms.
3) Please describe how sample size was arrived at. I think that sample size was small. To obtain transparency, it is necessary to explain the estimated sample size and the final sample size. I agree with the authors points on why sample size did not arrive at the estimated one, as shown in the limitation section.
4) In the "mediator analysis" section, I cannot understant how resilency can be a mediator of the association between stress and depresson, and between traumatic child birth perception and depression by this statistical analysis. Were all the analyses unadjusted measures?? Please explain any statistical analysis plan on how to control for potential confounders. I think it is impossible to assess the assumption of "traumatic childbirth perception, and pandemic-related pregnancy stress are ndependently associated with depression symptoms among pregnant women during the Covid-19 lockdown", and "the associations between pandemic-related stress and prenatal depression, as well as traumatic childbirth perception, are mediated through resiliency." only by this anslysis.
-minor
1) Please describe how missing data was addressed and analyzed.
2)Please avoid describing the results of previous studies in the results section to avoid misleadings. These references should be moved to discussion sections.
3) Previolus studies indicated that women suffering from depression in general (Skrove et al., 433 2013; Wu et al., 2020), in the postpartum period (Hain et al., 2016; Mautner et al., 2022), and antenatally as well (Nie et al., 2017; Zhang et al., 2020) have lower resilience levels. I think that previous studies have already assessed the influence of resilience on depression in women during the COVID-19 pandemic. Please review the previous studies and emphasize the novelty of this study.
I think it is necessary to revise the manuscript.
Author Response
Thank you for giving me the opportunity to review this manuscript.
This cross-sectional study describes potential impact of resilience on depression in Polish women with pregnancy during the COVID-19 pandemic.
I think it is necessary to revise the manuscript.
ANSWER: We appreciate the time you spend preparing a review. Thank you for all the valuable comments that increase the scientific level of the article.
All changes made to the text are marked in blue
-major
- Please describe the study design by using the PECO format. The Population is the Polish participants with pregnancy. The exposure is low resiliency. The control is high resiliency. The primary outcome is depressive symptoms. Please delete the sentences that the resiliency is the mediator.
ANSWER: Thank you for your attention and for pointing to the PECO format. As PECO was developed for systematic reviews, especially those involving exposure (see Morgan et al., 2018; https://doi.org/10.1016/j.envint.2018.07.015), we were unable to address all four of PECO's questions, but we included additional information based on PECO's assumptions into the Study design section.
We have thoroughly analyzed the Reviewer's recommendation, and it seems to us that we probably did not clearly describe the research assumptions, so we would like to thank you for this comment. Indeed prenatal depressive symptoms are the primary outcome. However, it is not that low resiliency means exposure, while high resilience means control group. We did not assign to the group based on the previously estimated level of resilience. The division into low and high was made at the level of statistical analysis. From the very beginning, we planned to conduct a mediation analysis to see if this variable changes the relationship between pandemic-related pregnancy stress symptoms and prenatal depressive symptoms (model 1) and traumatic childbirth perception and prenatal depressive symptoms (model 2). Referring to other studies (i.e. Zeng et al., 2022; https://doi.org/10.1186/s12884-022-04811-y), we assumed that resilience should mediate between pandemic-related pregnancy stress symptoms and prenatal depressive symptoms, and between traumatic childbirth perception and prenatal depressive symptoms. However, we wanted to check whether resilience still plays its protective role, i.e. whether it reduces or eliminates the negative impact of, for example, pregnancy stress on symptoms of depression in a different situation than previously studied. Different because it is related to the new, unknown and unexpected threat, the coronavirus, and the limitations resulting from functioning in a pandemic reality. Importantly, the term "mediator" refers to the statistical model - a mediator variable (resilience in our study) is the variable that causes mediation in the dependent and the independent variables. Thus, it explains the relationship between the dependent and independent variables.
We hope that after this explanation and completing the Study design paragraph, the assumptions of our study are clearer.
3) Please describe how sample size was arrived at. I think that sample size was small. To obtain transparency, it is necessary to explain the estimated sample size and the final sample size. I agree with the authors points on why sample size did not arrive at the estimated one, as shown in the limitation section.
ANSWER: Thank you for this comment, we understand the Reviewer's point of view. Indeed, the sample size is small. However, we want to clarify that the minimum sample size for the regression analysis (the mediation analysis is based on the assumptions of the regression analysis) was reached, and that was 55 subjects. We were unable to get a larger sample than 80 pregnant women for reasons described in the limitation section. In addition, post hoc, we checked the power of the analyzes carried out, and it was satisfactory.
The Data analysis section has been supplemented:
An a priori power analysis using G*Power indicated that a total sample size of n = 55 was required to provide 80% statistical power with α=.05 and detect a medium (f2=.15) effect size for two predictors.
2) Please describe what is potential confounders, and any effect modifiers. It is impossible to assess how the potential confounders affect the outcomes. For example, please describe unadjusted estimates and their precision (Model 1), those after adjusting for baseline characteristics (Model 2), and those after adjusting for baseline characteristics + potential confounders (stress scales and so on) (Model 3). Please make clear what kind of confounders was adjusted for and why they were included. Actually, table 4 showed that depressive scores were correlated with pandemic-related pregnancy stress scales and traumatic child blrth perception. I think that this result indicated that with pandemic-related pregnancy stress and traumatic child birth perception would be potential confounders of the association between resilience and depressive symptoms.
4) In the "mediator analysis" section, I cannot understant how resilency can be a mediator of the association between stress and depresson, and between traumatic child birth perception and depression by this statistical analysis. Were all the analyses unadjusted measures?? Please explain any statistical analysis plan on how to control for potential confounders. I think it is impossible to assess the assumption of "traumatic childbirth perception, and pandemic-related pregnancy stress are ndependently associated with depression symptoms among pregnant women during the Covid-19 lockdown", and "the associations between pandemic-related stress and prenatal depression, as well as traumatic childbirth perception, are mediated through resiliency." only by this anslysis.
ANSWER: Thank you. We will address comments 2 and 4 together.
The mediation models were built on the basis of theory and previous research. We do not apply for it only on the basis of the statistical analysis itself, but it has a substantive justification. Earlier studies on the role of resilience in maintaining mental health suggest that it may reduce or eliminate the impact of, for example, pandemic-related pregnancy stress on symptoms of depression in pregnancy. Previous studies have shown that the higher the stress level experienced during pregnancy, the more severe the symptoms of depression (or the greater the likelihood of developing symptoms of depression). There is also such a relationship in our study. The higher the scores on the pandemic-related pregnancy stress scale, the higher the scores on the scale for measuring depression. However, when we introduce resilience as a mediator, this relationship is statistically significant, i.e. stress ceases to affect the severity of depressive symptoms. We are talking then about full mediation. Our dependent variable is prenatal depression, and independent variables (considered separately in two models) are pandemic-related pregnancy stress and traumatic childbirth perception. Resilience is a mediator variable. This is how our model is built. We have not tested a model where stress or traumatic childbirth perception was to be the confounders between resilience and depressive symptoms. In the introduction, we write about the results of earlier research, which are the theoretical basis for our research model.
In this model, we did not control the impact of other variables, such as sociodemographic variables, as the potential confounders, although, of course, one should bear in mind their potential role. We address this in the Limitations section.
1) Please describe how missing data was addressed and analyzed.
ANSWER: Thank you for this important comment. There was no need to analyze the missing data as we only included complete questionnaires in the final analysis. As described in the Procedure and data collection section, 13 questionnaires were incomplete. However, these were not missing data in individual questions or items of the questionnaires, but the subjects, after completing the sociodemographic and/or gynaecological-obstetrics part of the whole survey, did not start the proper study part at all, so they were not included in the analyzes. All other questionnaires were complete with no missing data.
We provide an additional explanation in the Procedure and data collection section.
2)Please avoid describing the results of previous studies in the results section to avoid misleadings. These references should be moved to discussion sections.
ANSWER: Thank you for this comment. Indeed, references should not dominate the Results section. As suggested, most of the sentences in this section have been moved to the Discussions. We have only decided to leave a reference for the prenatal depression results. The references here are cited to indicate which criterion was used for the cut-off scores. It seems to us that it is more substantive to leave these references here.
3) Previolus studies indicated that women suffering from depression in general (Skrove et al., 433 2013; Wu et al., 2020), in the postpartum period (Hain et al., 2016; Mautner et al., 2022), and antenatally as well (Nie et al., 2017; Zhang et al., 2020) have lower resilience levels. I think that previous studies have already assessed the influence of resilience on depression in women during the COVID-19 pandemic. Please review the previous studies and emphasize the novelty of this study.
ANSWER: Thank you, this is a very valuable comment. Indeed, the relationship between depression and resilience has been studied before. Our assumptions are based on the results of previous studies, which show that women (both pregnant and those who have recently had a baby) with low resilience have higher scores on scales that measure depression. However, the studies we refer to were not conducted during the pandemic. This also applies to publications that appeared after the outbreak of the coronavirus pandemic (e.g. studies by Mautner et al., 2022 - carried out from December 2016 until December 2018, while studies by Zhang et al., 2020 - between July 2018 and July 2019). The novelty of our study is also the measurement tools. First, we used a scale to measure the stress typically associated with experiencing pregnancy during the coronavirus pandemic. Second, we used a new tool to measure traumatic childbirth perception (TPBS). It is a tool that allows you to determine the level of perceiving childbirth as traumatic not only in women who have given birth and may have negative memories of the course of childbirth. TPBS allows for the exploration of general attitudes and perceptions about childbirth. We assumed that it is important, especially in the period of a pandemic, when restrictions related to the course of childbirth have been introduced, especially regarding the presence of a partner by the mother in labour. Finally, we did not only check the differences in the severity of depressive symptoms depending on low and high resilience but a more complex relationship: whether resilience could change (reduce or eliminate) the relationship between stress and traumatic perception of childbirth and depression.
The Strengths and Limitations section has supplemented the manuscript with this information.
Reviewer 3 Report
Dear authors,
it was a great pleasure to read your manuscript.
Article is very interesting and well-organised.
I would like to ask you to consider to add the following citations to Introduction or Discussion section:
1. https://www.termedia.pl/Changes-in-maternity-care-in-Poland-perceived-r-nby-midwives-working-in-the-SARS-CoV-2-pandemic-r-nA-preliminary-study,134,46922,1,1.html
2. https://journals.viamedica.pl/ginekologia_polska/article/view/71111
3. https://www.mdpi.com/1660-4601/19/1/180
Author Response
ANSWER: Dear Reviewer, we appreciate the time you take to read the manuscript.
Thank you for pointing to the additional articles. All of them have been included in the Introduction and Discussion part of our article. All changes made to the text are marked in blue.
Reviewer 4 Report
This paper focuses on an important issue and should be considered for inclusion in this special issue. The Introduction is well written and referenced with current citations. The hypotheses are clearly stated and testable. The Inclusion/Exclusion criteria are very specific. My "problem" is with the study sample. Any woman interested in participating could send an email so stating and then given a link to the survey. The results are self-reports. Clearly there is an issue of selection bias (acknowledged by the authors in the Limitations section). The 80 women in the sample are well educated, urban, married, and financially well off (compared to those in lower SES categories). As such, the generalizability of the findings are low to moderate.
The validated scales used are clearly explained and the variables tested also clearly delineated. The data are well presented. However, I would like to see data stratified by demographic factors rather than just by presenting overall results. There probably would not be much difference since the sample is overwhelmingly homogenous, but it would have been interesting to see if there are differences in scores by select demographic factors. For example, is there a difference among those with previous pregnancies compared to those having their first child? There could theoretically be differences in scores by demographic factors, which should be reported. Who is more likely to score higher (or lower)?
As long as the authors clearly state how limited the findings are, I have no disagreement with the paper being published. Pity that the study could not be more rigorous. I personally am suspect of findings from these type of studies because they are methodologically flawed. Selection bias must be addressed more forcefully in this paper and the authors should stress that additional studies should be done to further understand the correlates of depression etc on pregnant women.
Author Response
Dear Reviewer, we appreciate the time you spend preparing a review. Thank you for all the valuable comments that increase the scientific level of the article.
All changes made to the text are marked in blue
This paper focuses on an important issue and should be considered for inclusion in this special issue. The Introduction is well written and referenced with current citations. The hypotheses are clearly stated and testable. The Inclusion/Exclusion criteria are very specific. My "problem" is with the study sample. Any woman interested in participating could send an email so stating and then given a link to the survey. The results are self-reports. Clearly there is an issue of selection bias (acknowledged by the authors in the Limitations section). The 80 women in the sample are well educated, urban, married, and financially well off (compared to those in lower SES categories). As such, the generalizability of the findings are low to moderate.
ANSWER: Thank you for this comment. We are aware of the limitations of the study, especially those relating to the study group. We made some additional comments in the Limitations section.
The validated scales used are clearly explained and the variables tested also clearly delineated. The data are well presented. However, I would like to see data stratified by demographic factors rather than just by presenting overall results. There probably would not be much difference since the sample is overwhelmingly homogenous, but it would have been interesting to see if there are differences in scores by select demographic factors. For example, is there a difference among those with previous pregnancies compared to those having their first child? There could theoretically be differences in scores by demographic factors, which should be reported. Who is more likely to score higher (or lower)?
ANSWER: Thank you, this is a very valuable comment. We checked prenatal depression scores for sociodemographic factors. There are no statistically significant differences, but we found an interesting difference - healthy pregnant women scored higher, not high-risk pregnancies, as we might expect. As mentioned above, the difference is not statistically significant, but worth noting and commenting on what we did. A table (Table 2) showing the detailed results has been added to the manuscript.
As long as the authors clearly state how limited the findings are, I have no disagreement with the paper being published. Pity that the study could not be more rigorous. I personally am suspect of findings from these type of studies because they are methodologically flawed. Selection bias must be addressed more forcefully in this paper and the authors should stress that additional studies should be done to further understand the correlates of depression etc on pregnant women.
ANSWER: The limitations section has been supplemented with additional information on selection bias and recommendations for further research.
Reviewer 5 Report
Thank you for giving me to review your manuscript. This manuscript is interesting and scientifically meaningful for considering the mediating effect of resiliency on prenatal depression symptoms in the COVID-19 pandemic. Regarding the contents, the following revision should be considered.
The title should include the study design.
There are long paragraphs. The author should focus on theory building, the problems, and research question paragraphs. The first to third paragraphs contain mixed contents. The first paragraph should focus on the general information between pregnancy and mental conditions. Moreover, the second and third paragraphs should introduce the effect of COVID-19 as the theoretical and conceptual framework, including research questions.
The introduction should clearly include the research question of this study.
In the introduction, the contents focused on developed countries and urban contexts. The COVID-19 pandemic impinged on rural contexts as well. The authors should describe the situation of the pandemic by using the following articles.
- Apaijitt, P. and V. Wiwanitkit, Knowledge of coronavirus disease 2019 (COVID-19) by medical personnel in a rural area of Thailand. Infection Control & Hospital Epidemiology, 2020: p. 1-1.
- Ogunkola, I.O., et al., Rural communities in Africa should not be forgotten in responses to COVID-19. Int J Health Plann Manage, 2020. 35(6): p. 1302-1305.
- Takashima, R.; Onishi, R.; Saeki, K.; Hirano, M. Perception of COVID-19 Restrictions on Daily Life among Japanese Older Adults: A Qualitative Focus Group Study. Healthcare 2020, 8, 450.
- Ohta, R., Y. Ryu, and C. Sano, Fears Related to COVID-19 among Rural Older People in Japan. Healthcare (Basel), 2021. 9(5).
- Ranscombe, P., Rural areas at risk during COVID-19 pandemic. Lancet Infect Dis, 2020. 20(5): p. 545.
The sample section of the method contains no descriptions regarding sample calculation.
The discussion part should be based on paragraph writing. There are too long paragraphs, and they are not friendly for readers.
This study should describe the limitation of sampling bias and the results' applicability to other settings, and the future investigation in the limitation part.
In the conclusion or discussion, the study’s strengths should be focused on international readers.
Author Response
Thank you for giving me to review your manuscript. This manuscript is interesting and scientifically meaningful for considering the mediating effect of resiliency on prenatal depression symptoms in the COVID-19 pandemic.
ANSWER: We appreciate the time you spend preparing a review. Thank you for all the valuable comments that increase the scientific level of the article.
Regarding the contents, the following revision should be considered
The title should include the study design.
ANSWER: The title has been changed to include information about the study design
There are long paragraphs. The author should focus on theory building, the problems, and research question paragraphs. The first to third paragraphs contain mixed contents. The first paragraph should focus on the general information between pregnancy and mental conditions. Moreover, the second and third paragraphs should introduce the effect of COVID-19 as the theoretical and conceptual framework, including research questions.
ANSWER: Thank you for your comment. We have made changes in the Introduction part.
The introduction should clearly include the research question of this study.
ANSWER: Research questions have been added to the Introduction section
In the introduction, the contents focused on developed countries and urban contexts. The COVID-19 pandemic impinged on rural contexts as well. The authors should describe the situation of the pandemic by using the following articles.
- Apaijitt, P. and V. Wiwanitkit, Knowledge of coronavirus disease 2019 (COVID-19) by medical personnel in a rural area of Thailand. Infection Control & Hospital Epidemiology, 2020: p. 1-1.
- Ogunkola, I.O., et al., Rural communities in Africa should not be forgotten in responses to COVID-19. Int J Health Plann Manage, 2020. 35(6): p. 1302-1305.
- Takashima, R.; Onishi, R.; Saeki, K.; Hirano, M. Perception of COVID-19 Restrictions on Daily Life among Japanese Older Adults: A Qualitative Focus Group Study. Healthcare 2020, 8, 450.
- Ohta, R., Y. Ryu, and C. Sano, Fears Related to COVID-19 among Rural Older People in Japan. Healthcare (Basel), 2021. 9(5).
- Ranscombe, P., Rural areas at risk during COVID-19 pandemic. Lancet Infect Dis, 2020. 20(5): p. 545
ANSWER: Thank you for pointing to the additional articles. All of them have been included in our article in the Introduction part.
The sample section of the method contains no descriptions regarding sample calculation.
ANSWER: Thank you. The following information was included:
An a priori power analysis using G*Power indicated that a total sample size of n = 55 was required to provide 80% statistical power with α=.05 and detect a medium (f2=.15) effect size for two predictors.
The discussion part should be based on paragraph writing. There are too long paragraphs, and they are not friendly for readers.
ANSWER: Suggested changes were provided.
This study should describe the limitation of sampling bias and the results' applicability to other settings, and the future investigation in the limitation part.
ANSWER: The limitations section has been supplemented with additional information on selection bias and recommendations for further research.
In the conclusion or discussion, the study’s strengths should be focused on international readers.
ANSWER: The strengths of the study were included. We added the strengths to the Strength and Limitations section. Thank you for this comment.
Round 2
Reviewer 1 Report
Thanks. I have no further comments. Good luck!
Reviewer 2 Report
Thank you so much for revising the manuscript.
I think this manuscript would be suitable for publication in this journal.
Reviewer 5 Report
The manuscript has been considerably improved. I think that this paper is suited for inclusion in our journal.